**www.cambridge.org/qrd**

## Perspective

**Key words:**
biomolecular recognition; enhanced sampling; kinetics; machine learning; molecular dynamics; thermodynamics

**Author for correspondence:**
*Yinglong Miao,
E-mail: miao@ku.edu

**CAMBRIDGE UNIVERSITY PRESS**

# Challenges and frontiers of computational modelling of biomolecular recognition

Jinan Wang [ID], Apurba Bhattarai, Hung N. Do [ID] and Yinglong Miao* [ID]

Center for Computational Biology and Department of Molecular Biosciences, University of Kansas, Lawrence, KS 66047, USA

## Abstract

Biomolecular recognition including binding of small molecules, peptides and proteins to their target receptors plays a key role in cellular function and has been targeted for therapeutic drug design. However, the high flexibility of biomolecules and slow binding and dissociation processes have presented challenges for computational modelling. Here, we review the challenges and computational approaches developed to characterise biomolecular binding, including molecular docking, molecular dynamics simulations (especially enhanced sampling) and machine learning. Further improvements are still needed in order to accurately and efficiently characterise binding structures, mechanisms, thermodynamics and kinetics of biomolecules in the future.

## Introduction

Biomolecular recognition plays key roles in many fundamental biological processes, including immune response, cellular signal transduction and so on (Nooren and Thornton, 2003). Moreover, these processes are implicated in the development of numerous human diseases and serve as important drug targets (Ferreira *et al.*, 2016; Scott *et al.*, 2016). Experimental techniques (Miura, 2018) including X-ray crystallography, nuclear magnetic resonance (NMR) and cryo-electron microscopy (cryo-EM) have been applied to determine the bound structures of protein–small molecule, protein–peptide and protein–protein complexes. The number of experimental complex structures are significantly increased in recent years (Sussman *et al.*, 1998). However, it is still rather time consuming and resource demanding to obtain high-resolution experimental structures. Moreover, the experimental structures often capture static pictures of protein complexes. Intermediate conformational states that could be relevant for drug design are usually difficult to probe using current experimental techniques.

Computational methods have been developed to model biomolecular recognition and predict the binding free energies and/or kinetic rates, including the widely used molecular docking (Morris *et al.*, 2009; Wang and Zhu, 2016; Porter *et al.*, 2017; Ciemny *et al.*, 2018; Vakser, 2020), Brownian dynamics (Ermak and McCammon, 1978; Gabdoulline and Wade, 2001; Spaar *et al.*, 2006; Wieczorek and Zielenkiewicz, 2008; Votapka and Amaro, 2015) and molecular dynamics (MD) simulations (Karplus and McCammon, 2002; Basdevant *et al.*, 2013; Pan *et al.*, 2019; He *et al.*, 2021; Lamprakis *et al.*, 2021). Molecular docking has been widely used for predicting the *holo* structures of protein–ligand (Wang and Zhu, 2016), protein–peptide (Ciemny *et al.*, 2018) and protein–protein complexes (Vakser, 2020). Although significant improvements have been achieved in developments of the molecular docking algorithms, the accuracy of docking could be still limited, due to high system flexibility especially in docking of the peptides and proteins. Recently, deep learning techniques have been introduced into molecular docking to increase accuracy. One successful example is the AlphaFold-multimer (Evans *et al.*, 2022), which has significantly increased the accuracy of predicting protein–protein complex structures. However, one is still not able to predict biomolecular binding kinetics from molecular docking.

MD is a powerful technique for simulations of biomolecular structural dynamics (Karplus and McCammon, 2002). Remarkable advances in computing hardware (e.g., the Anton supercomputer and GPUs) and software developments have significantly increased the accessible time scale of conventional MD (cMD) from hundreds of nanoseconds to hundreds of microseconds (Harvey *et al.*, 2009; Shaw *et al.*, 2010; Johnston and Filizola, 2011; Lane *et al.*, 2013; Hollingsworth and Dror, 2018; Shaw *et al.*, 2021). Notably, the latest Anton3 (Shaw *et al.*, 2021) has achieved the speed of hundreds of microseconds per day for ATPase and Satellite Tobacco Mosaic Virus (STMV) with total number of atoms ranging from 328 K to 1,067 K, which will significantly facilitate simulations of biomolecular recognition process. The cMD simulations have been widely applied to investigate biomolecular dynamics, including conformational change (Jensen *et al.*, 2012), protein folding (Lindorff-Larsen *et al.*, 2011) and substrate binding (Shan *et al.*, 2011; Dror *et al.*, 2013; Robustelli *et al.*, 2020).

For small-molecule ligand binding, Shan *et al.* (2011) observed spontaneous binding of the Dasatinib drug to its target Src kinase during tens of microseconds cMD simulations. However, no dissociation event was observed in the cMD simulations. Pan *et al.* (2017) performed tens of microseconds cMD simulations to successfully characterise repetitive binding and dissociation of six small-molecule fragments to the protein FKBP. Based on the large number of binding and dissociation events in the simulations, they were able to accurately calculate the binding free energies and kinetic rates. Remarkably, the binding free energies calculated from the cMD simulations agreed very well with those predicted from free energy perturbations (FEP) calculations. It is worth noting that the tested fragments were weak binders with affinities ranging from 200 μM to 20 mM. It is still challenging to simulate both binding and dissociation of typical small-molecule ligands of proteins (usually with higher binding affinities and slower dissociation rates) using cMD, although the ligand residence time (or dissociation rate) has recently been recognised to correlate better with drug efficacy (Schuetz *et al.*, 2017). For protein–protein interactions, tens of microseconds cMD simulations were able to capture barnase binding to barstar (Pan *et al.*, 2019). Accurate barnase binding rate ($k_{on}$) was predicted based on multiple binding events captured in a total of ~440 μs Anton cMD simulations (Pan *et al.*, 2019). However, it remains challenging to simulate dissociation of the barnase–barstar model system using cMD (Pan *et al.*, 2019).

Weighted ensemble (Saglam and Chong, 2019) and Markov state model (MSM) (Plattner *et al.*, 2017) have been developed to improve the prediction of biomolecular binding thermodynamics and kinetics based on a large number of short cMD trajectories. The kinetic binding rate ($k_{on}$) of the p53 peptide to the MDM2 protein was accurately predicted with weighted ensemble of a total amount of ~120 μs cMD simulations in implicit solvent (Zwier *et al.*, 2016). Another weighted ensemble of a total of ~18 μs cMD simulations was able to accurately predict the barnase–barstar binding rate constant ($k_{on}$) (Saglam and Chong, 2019). However, it is still challenging to model the slow protein/peptide dissociation processes with weighted ensemble simulations (Zwier *et al.*, 2016; Saglam and Chong, 2019). MSM (Plattner and Noe, 2015; Paul *et al.*, 2017; Plattner *et al.*, 2017) was able to simultaneously predict the binding and dissociation kinetics through longer aggregated cMD simulations. MSM built with 150 μs MD simulation data was used to accurately predict benzamidine–trypsin binding kinetics (Plattner and Noe, 2015). Based on a total of two millisecond cMD simulations of barnase binding to barstar, MSM was generated to predict intermediate structures, binding energies and kinetic rates that were consistent with experimental data (Plattner *et al.*, 2017). However, these calculations required very expensive computational resources.

Coarse-grained MD models have been developed to reduce the demand of computational resources and extend simulation time scales (Souza *et al.*, 2020, 2021). Souza *et al.* (2020) performed millisecond cMD simulations to capture the binding of diverse protein–ligand systems. Accurate binding free energies were predicted through the cMD simulations without *a priori* information (Souza *et al.*, 2020). Millisecond MD simulations with a useful coarse-grained model (PACE) were performed to characterise the binding mechanism of the intrinsically disordered Aβ peptides (Aβ$_{17–42}$) to form Aβ fibril (Han and Schulten, 2014). In addition, coarse-grained models could be incorporated into multiscale computational approaches to improve the efficiency and accuracy of ligand binding thermodynamics and kinetics calculations (Elber, 2020; Jagger *et al.*, 2020; Huang, 2021). For example, simulation

enabled estimation of kinetic rates (SEEKR; Votapka and Amaro, 2015; Jagger *et al.*, 2020) is a multiscale simulation approach combining MD, Brownian dynamics and milestoning for calculating receptor–ligand binding and dissociation rates. SEEKR has been shown to estimate binding kinetic rates with up to a factor of 10 less simulation time (Jagger *et al.*, 2020).

Enhanced sampling methods have been developed to efficiently simulate biomolecular recognition. They could be generally divided into two categories depending on the usage of collective variables (CVs). The CV-based methods include the widely used steered MD (Kingsley *et al.*, 2016), umbrella sampling (Gumbart *et al.*, 2013; Kingsley *et al.*, 2016; Joshi and Lin, 2019b), metadynamics (Antoszewski *et al.*, 2020; Banerjee and Bagchi, 2020), adaptive biasing force (ABF; Darve and Pohorille, 2001; Darve *et al.*, 2008) and so on. These methods often use predefined CVs to effectively guide simulations. Thus, *a priori* knowledge of the system is required in CV-based enhanced sampling. Alternatively, when it is difficult to predefine CVs, CV-free enhanced sampling methods could be useful (Kamenik *et al.*, 2022). These methods include replica exchange MD (Sugita and Okamoto, 1999; Sugita *et al.*, 2019; Siebenmorgen and Zacharias, 2020), random acceleration molecular dynamics (RAMD; Nunes-Alves *et al.*, 2021), tempered binding (Pan *et al.*, 2019), integrated tempering sampling (ITS; Yang *et al.*, 2015; Shao and Zhu, 2019), scaled MD (Deb and Frank, 2019), accelerated MD (aMD; Hamelberg *et al.*, 2004), Gaussian accelerated MD (GaMD; Miao *et al.*, 2015b; Wang *et al.*, 2021) and so on. The above-mentioned methodological advances have enabled simulations of millisecond or even longer time scale processes. Here, we will briefly review recent efforts in modelling biomolecular recognition, especially characterisation of binding thermodynamics and kinetics.

## Collective variable-based enhanced sampling

During CV-based enhanced sampling simulations, a potential or force bias is applied along certain CVs to facilitate energy barrier crossing events among different conformational states. Typical CVs include distances, angle, dihedral, path, eigenvectors generated from the principal component analysis, root-mean square deviation (RMSD) relative to a reference conformation (Bouvier and Grubmuller, 2007) and so on. The bias potential applied to the system is usually around several kcal/mol. Thus one is able to accurately recover the original free energy profiles.

Umbrella sampling has been applied to predict the ligand/peptide/protein binding and/or dissociation pathways and map the associated free energy landscapes (Gumbart *et al.*, 2013; Joshi and Lin, 2019a; Sieker *et al.*, 2008; You *et al.*, 2019). Metadynamics has been applied to investigate ligand/peptide/protein binding in terms of the binding kinetic rates (Casasnovas *et al.*, 2017; Sun *et al.*, 2017) and free energies (Saleh *et al.*, 2017; Banerjee *et al.*, 2018; Raniolo and Limongelli, 2020; Wang *et al.*, 2022a). Metadynamics simulations (Limongelli *et al.*, 2013; Tiwary and Parrinello, 2013) have also been applied to investigate the thermodynamics and kinetics of benzamidine inhibitor binding to trypsin. Multiple metadynamics trajectories with a total of 5 μs simulations were obtained to predict the ligand unbinding pathways and dissociation rate constant ($k_{off}$). The predicted $k_{off}$ (9.1 ± 2.5 s$^{-1}$) was smaller than the experimental value (600 ± 300 s$^{-1}$). Separate funnel metadynamics simulations predicted accurate of ligand binding free energies (−8.5 ± 0.7 kcal/mol) for the same system

(Limongelli *et al.*, 2013). Infrequent metadynamics simulations with three carefully chosen CVs have successfully predicted the peptide binding and dissociation rates for the system of p53-MDM2 -(Zou *et al.*, 2020). Although these methods have shown remarkable improvements in capturing rare events that happen over exceedingly long timescales, users often face a challenge for defining CVs, which requires expert knowledge of the studied systems (Abrams and Bussi, 2014; Zuckerman, 2011). Additionally, the predefined CVs could constrain the sampling space, leading to slow convergence of the simulations and suffering from "hidden energy barrier" once important CVs were missed during the simulation setup (Bešker and Gervasio, 2012). To accelerate the convergence of simulations, replica exchange or parallel tempering methods have been incorporated into metadynamics. For example, bias-exchange metadynamics simulations with eight CVs have been performed to predict accurate binding free energy of the p53 peptide to the MDM2 protein. Parallel tempering metadynamics simulations with well-tempered ensemble (PTMetaD-WTE) successfully captured the binding and dissociation processes of insulin dimer (Antoszewski *et al.*, 2020). In summary, by carefully defining reaction coordinates, the CV-based enhanced sampling methods could efficiently and accurately predict binding free energies and kinetic rates.

### Enhanced sampling without predefined collective variables

In CV-free enhanced sampling methods, bias is often applied on generalised properties of the system (such as the potential energy and atomic forces) in the simulations. Repetitive benzamidine binding and unbinding in trypsin were captured using the selective ITS method (Yang and Qin Gao, 2009; Yang *et al.*, 2015; Shao and Zhu, 2019). Pan *et al.* (2019) developed the tempered binding method, which significantly accelerates the slow protein dissociation process by dynamically adjusting electrostatic and van der Waals interactions between different groups of protein atoms by a factor λ. The tempered binding simulations have successfully captured repetitive binding and dissociation events for five diverse protein–protein systems (Pan *et al.*, 2019). In the scaled MD simulations (Sinko *et al.*, 2013), a scale factor ranging from 0 to 1 is introduced to smoothen the potential energy surface. Schuetz *et al.* (2018) performed scaled MD simulations to accurately predict the residence time and drug dissociation pathways of different inhibitors of heat shock protein 90 (Hsp90). In a recent study, Bianciotto *et al.* (2021) used scaled MD simulations to predict the residence time and ligand unbinding pathways for a set of 27 ligands of Hsp90 protein, being highly consistent with experimental data. Deb and Frank (2019) developed a selective scaled MD simulation method, where specific energy terms are scaled to promote dissociation of bound ligands from the protein. Particularly, ligand–water interactions are scaled to help the ligands dissociate from its bound state. Selective scaled MD predict accurate residence times and associated free energy change of three inhibitor drugs bound to cyclin-dependent kinase protein complexes. Hence, selective scaled MD proves to be an important enhanced sampling method for modelling biomolecular dissociation process.

In RAMD, an additional random force is applied on the ligand to promote especially the dissociation. In one recent study, Nunes-Alves *et al.* (2021) performed RAMD simulations to predict ligand dissociation rates of T4 lysozyme. The predicted kinetic rates agreed well with experimental values for various systems with different ligands, temperatures and protein mutations. Moreover,

a ligand with complex dissociation pathways was often associated with longer residence time. In another study, the same group (Kokh and Wade, 2021) performed RAMD simulations to explore ligand dissociation pathways and kinetics of two GPCRs, i.e., the $\beta_2$ adrenergic receptor ($\beta_2$AR) and M2 muscarinic acetylcholine receptor (M2R). The ligand dissociation pathways observed in the RAMD simulations were similar to those in long cMD and metadynamics simulations. Additionally, RAMD revealed an allosteric modulation mechanism of the LY2119620 PAM in the M2R. Dissociation of the iperoxo agonist was blocked from one of the possible pathways and hence had increased residence time, being consistent with the experimental data.

The aMD enhanced sampling technique works by adding a non-negative boost potential to smooth the system potential energy surface (Voter, 1997; Hamelberg *et al.*, 2004). The boost potential ($\Delta V$) decreases the energy barrier to facilitate the system cross different conformational states (Hamelberg *et al.*, 2004, 2007). In one study, Kappel *et al.* (2015) performed aMD simulations to study ligand binding to M3 muscarinic receptor (M3R). Three ligands of the receptor: full agonist Ach, partial agonist arecoline (Arc) and antagonist tiotropium (TTP) were used to perform the aMD simulations. Starting from the bulk solvent, aMD captured the binding of Ach to the M3R orthosteric site in significantly less time as compared to the cMD simulations. The Arc was also observed binding to the orthosteric site whereas the TTP molecule bound to the extracellular vestibule of the receptor. Moreover, all ligands could bind to the extracellular vestibule of the receptor, suggesting the vestibule as metastable binding site for orthosteric ligands. However, aMD suffers from large energetic noise during reweighting as the boost potential is typically on the order of tens to hundreds of kcal/mol (Shen and Hamelberg, 2008).

GaMD is developed to apply a harmonic boost potential to enhance sampling with significantly reduced energetic noise. The boost potential normally exhibits a near Gaussian distribution, which enables proper reweighting of the free energy profiles through cumulant expansion to the second order (Miao *et al.*, 2015b; Wang *et al.*, 2021). GaMD has been successfully applied to simulate important biomolecular processes, including ligand/protein/RNA binding (Miao *et al.*, 2015a, 2018b; Miao and McCammon, 2016; Pang *et al.*, 2017; Wang and Chan, 2017; Chuang *et al.*, 2018; Liao and Wang, 2019; Wang *et al.*, 2022b), protein folding (Miao *et al.*, 2015a; Pang *et al.*, 2017) and protein conformational changes (Miao and McCammon, 2016; Salawu, 2018; Zhang *et al.*, 2018). However, it remained challenging to simulate repetitive substrate binding and dissociation through normal GaMD (Miao and McCammon, 2018; Wang *et al.*, 2021).

Recently, "selective GaMD" algorithms have been developed to allow for more efficient enhanced sampling of biomolecular binding and dissociation processes, including the Ligand GaMD (LiGaMD) (Miao *et al.*, 2020), Peptide GaMD (Pep-GaMD; Wang and Miao, 2020) and protein–protein interaction – GaMD (PPI-GaMD; Wang and Miao, 2022). For simulations of biomolecular binding, the system contains substrate L (e.g. small-molecule ligands, peptides or ligand protein), protein $P$ and the biological environment $E$. Therefore, the potential energy of system could be decomposed into the following terms: $V(r) = V_{P,b}(r_P) + V_{L,b}(r_L) + V_{E,b}(r_E) + V_{PP,nb}(r_P) + V_{LL,nb}(r_L) + V_{EE,nb}(r_E) + V_{PL,nb}(r_{PL}) + V_{PE,nb}(r_{PE}) + V_{LE,nb}(r_{LE})$, where $V_{P,b}$, $V_{L,b}$ and $V_{E,b}$ are the bonded potential energies in protein $P$, substrate $L$ and environment $E$, respectively. $V_{PP,nb}$, $V_{LL,nb}$ and $V_{EE,nb}$ are the self non-bonded potential energies in protein $P$, substrate $L$ and environment $E$, respectively. $V_{PL,nb}$, $V_{PE,nb}$ and $V_{LE,nb}$ are the non-bonded

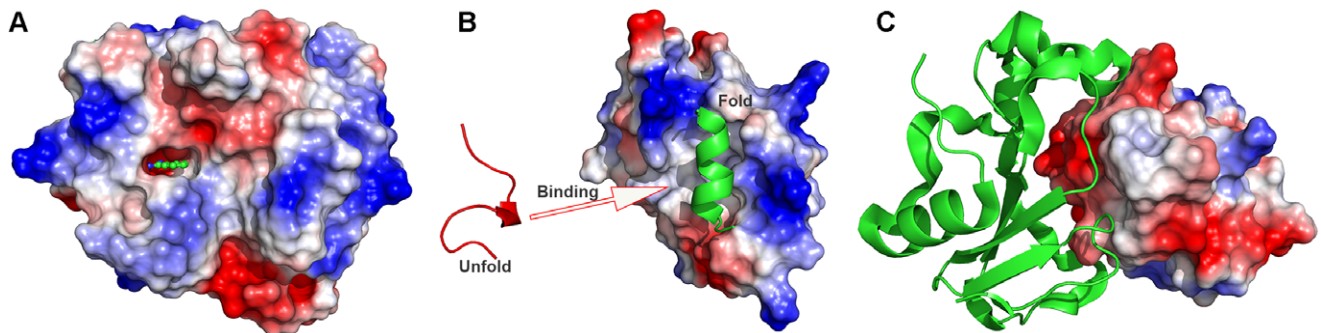

**Figure 1.** Schematic illustration of biomolecular recognition: (*a*) Small-molecule ligand binding, (*b*) peptide binding and (*c*) protein–protein interactions (PPIs).

interaction energies between *P-L*, *P-E* and *L-E*, respectively. In order to facilitate the ligand/peptide/protein binding (Fig. 1), a boost potential is selectively added on the essential energy terms ($V_{select}(r)$) in the LiGaMD, Pep-GaMD and PPI-GaMD, respectively. Presumably, ligand binding mainly involves the non-bonded interaction energies of the ligand. LiGaMD thus selectively boosts on the energy terms of $V_{select}(r) = V_{LL,nb}(r_L) + V_{PL,nb}(r_{PL}) + V_{LE,nb}(r_{LE})$. In comparison, peptide binding involves in both the bonded and non-bonded interaction energies of the peptide since peptides often undergo large conformational changes during binding to the target proteins. Thus, the essential energy term in Pep-GaMD is $V_{select}(r) = V_{LL,b}(r_L) + V_{LL,nb}(r_L) + V_{PL,nb}(r_{PL}) + V_{LE,nb}(r_{LE})$. While protein–protein binding and unbinding processes mainly involve the non-bonded interaction energies between protein partners, one can apply a selective boost to the essential energy term $V_{select}(r) = V_{PL,nb}$ in PPI-GaMD. In addition to selectively boost the essential energy term $V_{select}(r)$, another boost potential could be applied on the remaining energy of the system to facilitate substrate rebinding in a dual-boost scheme. These new algorithms have been implemented in the GPU version of AMBER22 (Case et al. 2022).

Repetitive binding and dissociation of small-molecule ligands were captured in the LiGaMD simulations of host–guest and protein–ligand binding model systems (Miao *et al.*, 2020), which enabled us to calculate ligand binding thermodynamics and kinetics calculations. Repetitive guest binding and dissociation in the β-cyclodextrin host were observed in hundreds-of-nanoseconds LiGaMD simulations. The binding free energies of guest molecules predicted from LiGaMD simulations agreed excellently with experimental data (< 1.0 kcal/mol error). In comparison with previous microsecond-timescale cMD simulations, accelerations of ligand kinetic rate constants in LiGaMD simulations were properly estimated using Kramers' rate theory. Furthermore, microsecond LiGaMD simulations observed repetitive benzamidine binding and dissociation in trypsin. Trypsin–benzamidine ligand binding free energy was calculated from the 3D PMF profile to be $-6.13 \pm 0.35$ kcal/mol, being highly consistent with the experimental value of $-6.2$ kcal/mol (Guillain and Thusius, 1970). Similarly, the ligand binding and dissociation time periods were recorded to calculate the reweighted $k_{on}$ and $k_{off}$ values to be $1.15 \pm 0.79 \times 10^7$ M$^{-1}\cdot$s$^{-1}$ and $3.53 \pm 1.41$ s$^{-1}$, respectively. These data were comparable to the values calculated from experiments (Guillain and Thusius, 1970).

Pep-GaMD (Wang and Miao, 2020) has been demonstrated on binding of three model peptides to the SH3 domains (Ball *et al.*, 2005; Ahmad and Helms, 2009), including "PAMPAR" (PDB: 1SSH),

"PPPALPPKK" (PDB: 1CKA) and "PPPVPPRR" (PDB: 1CKB). Repetitive dissociation and binding of the three peptides were successfully captured in each of the 1 microsecond Pep-GaMD simulations. The peptide binding free energies calculated from Pep-GaMD simulations were in excellent agreements with those from the experiments. For the 1CKA system, the calculated peptide binding free energy value was $-7.72 \pm 0.54$ kcal/mol, being highly consistent with the experimental value of $-7.84$ kcal/mol (Wu *et al.*, 1995). For the 1CKB system, the predicting binding free energy was $-6.84 \pm 0.14$ kcal/mol, being closely similar to the experimental value of $-7.24$ kcal/mol (Wu *et al.*, 1995). In addition, the Pep-GaMD predicted the $k_{on}$ and $k_{off}$ of 1CKA as $4.06 \pm 2.26 \times 10^{10}$ M$^{-1}\cdot$s$^{-1}$ and $1.45 \pm 1.07 \times 10^3$ s$^{-1}$, respectively. They were comparable to the experimental data (Xue *et al.*, 2014) of $k_{on}^{exp} = 1.5 \times 10^9$ M$^{-1}\cdot$s$^{-1}$ and $k_{off}^{exp} = 8.9 \times 10^3$ s$^{-1}$.

More recently, Pep-GaMD simulations were combined with complementary biochemical experiments to elucidate mechanism of tripeptide trimming of amyloid β-peptide (Aβ peptide) by γ-secretase (Bhattarai *et al.*, 2022). The active model of γ-secretase for ε cleavage was extracted from previous study (Bhattarai *et al.*, 2020) and used as the starting structure for Pep-GaMD simulations. 600 ns Pep-GaMD simulations were able to capture the ζ cleavage activation starting from the ε cleavage activated model, which was suggested to carry out in timescale of minutes (Kamp *et al.*, 2015). During activation, coordinated hydrogen bonds were formed between carbonyl oxygen of Aβ49 Val46 and enzyme catalytic Asp257. The two catalytic aspartates, Asp257 and Asp385 in the active site of the enzyme both formed hydrogen bonds with the water molecule aligned in between them. This activated enzyme conformation was well oriented for the ζ cleavage of amide between Val46 and Ile47 of the Aβ49. Three low energy states including "Final", "Intermediate" and "Initial" were identified from the Pep-GaMD simulations (Fig. 2*a*). The Final state denoted the activated enzyme conformation for ζ cleavage where the Asp257–Asp385 distance was ~7–8 Å and the Asp257–Aβ49 Val46 distance was ~3 Å (hydrogen bond). The Initial and Intermediate low energy states denoted the starting and transitional conformation during the activation process. Furthermore, Pep-GaMD simulations were performed for three additional FAD mutant Aβ49 bound enzyme systems. Similar to the wildtype system, Pep-GaMD simulations of I45F, A42T and V46F mutant Aβ49 bound enzyme systems were able to capture the ζ cleavage activation starting from the ε cleavage activated model. Free energy profiles of the FAD mutant systems were similar to the wildtype system (Fig. 2*b–d*). In the I45F mutant system, two low energy states

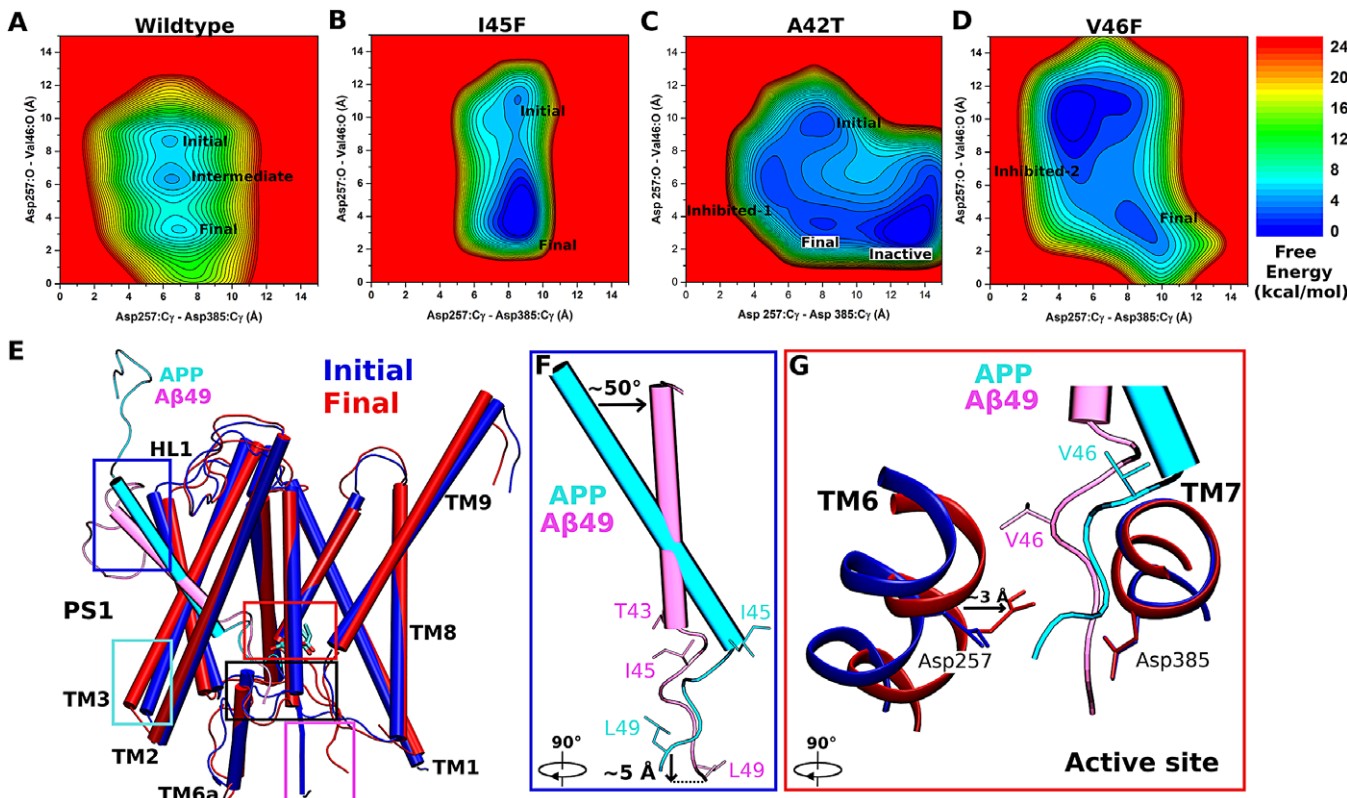

**Figure 2.** Mechanism of tripeptide trimming of amyloid β-peptide 49 by γ-secretase. 2D free energy profiles calculated regarding Asp257 - Asp 385 distance and Asp257 – Aβ49 Val46 distance calculated from Pep-GaMD simulations of (a) wildtype Aβ49 bound γ-secretase, (b) I45F mutant Aβ49 bound γ-secretase, (c) A42T mutant Aβ49 bound γ-secretase and (d) V46F mutant Aβ49 bound γ-secretase systems. (e) Structures of catalytic subunit PS1 bound to APP and Aβ49 substrates representing the "Initial" and "Final" conformational states, respectively. (f) Conformational changes in (f) Aβ49 and (g) active site of the enzyme during transition from Initial to Final activated state for ζ cleavage. Adapted with permission from Bhattari A, Devkota S, Do HN, Wang J, Bhattarai S, Wolfe MS and Miao Y. *Journal of the American Chemical Society.* 10.1021/jacs.1c10533. Copyright 2022 American Chemical Society.

were identified including "Initial" and "Final" (Fig. 2b). The A42T mutant was the most dynamic enzyme system with four distinct low energy states identified in a larger area covered free energy profile including "Initial", "Final", "Inhibited-1" and "Inactive" (Fig. 2c). The catalytic aspartates of the "Inhibited-1" conformational state were too close for activation and hence was inhibited. In contrast, the aspartates were too far for their catalytic activity in the "Inactive" low energy state of the enzyme. In the V46F mutant γ-secretase system, two low energy states were identified in the free energy profile including "Final" and "Inhibited-2" (Fig. 2d). The structures were compared between the "Initial" and "Final" low energy conformational states of the enzyme as identified from the free energy profiles (Fig. 2e–g). The enzyme moved from Initial to Final conformational state, the Aβ49 substrate tilted by ~50° (Fig. 2f). Unwinding of helix was observed in the C-terminus of Aβ49 where residues Val44 and Ile45 were observed changing their conformation from helix to a loop (Fig. 2f). Similarly, in the active site of the enzyme, the protonated Asp257 in the Final state was observed moving forward towards the substrate scissile amide bond by 3 Å in comparison to the Initial state (Fig. 2g). In contrast, the deprotonated Asp385 in the Final state and the Initial state were observed in a similar conformation (Fig. 2g). The simulation findings were highly consistent with biochemical experimental data. Taken together, complementary biochemical experiments and Pep-GaMD simulations have enabled elucidation of the mechanism of tripeptide trimming of Aβ49 by γ-secretase.

PPI-GaMD (Wang and Miao, 2022) has been demonstrated on a model system of the ribonuclease barnase binding to barstar. Six independent 2 μs PPI-GaMD simulations have successfully captured repetitive barstar dissociation and rebinding events (Fig. 3a). Five binding and six dissociation events were observed in both Sim1 and Sim3. In Sim2, three binding and four dissociation events were captured. For the remaining simulations (Sim4–Sim6), three binding and three dissociation events were observed (Fig. 3a). The barstar binding free energy predicted from PPI-GaMD was −17.79 kcal/mol with a standard deviation of 1.11 kcal/mol, being highly consistent with the experimental value of −18.90 kcal/mol (Schreiber and Fersht, 1993). In addition, the PPI-GaMD simulations allowed us to calculate the protein binding kinetics. The average reweighted $k_{on}$ and $k_{off}$ were predicted as $21.7 \pm 13.8 \times 10^8$ M$^{-1}$·s$^{-1}$ and $7.32 \pm 4.95 \times 10^{-6}$ s$^{-1}$, being highly consistent with the corresponding experimental values of $6.0 \times 10^8$ M$^{-1}$·s$^{-1}$ and $8.0 \times 10^{-6}$ s$^{-1}$, respectively. Furthermore, PPI-GaMD simulations have provided mechanistic insights into barstar binding to barnase, which involve long-range electrostatic interactions and multiple binding pathways (Fig. 3c–f), being consistent with previous experimental and computational findings of this model system. It is worth noting that at least three independent replicas of selective GaMD simulations with longer simulation lengths (e.g., microsecond) are required to obtain sufficient statistics for ligand binding, peptide binding and protein–protein interactions. In order to calculate accurate binding free energy and kinetic rates, the length of each simulation should be long enough to capture

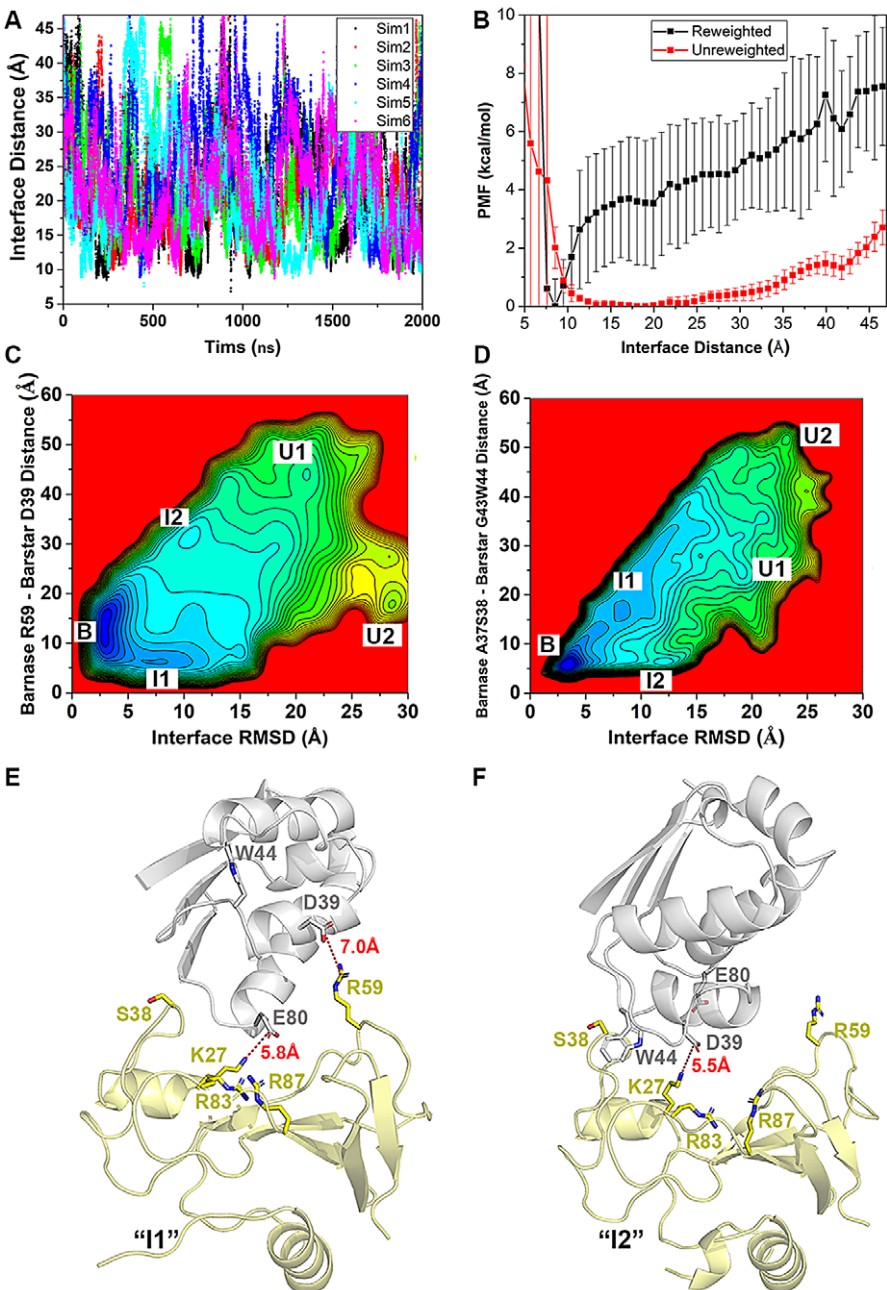

**Figure 3.** PPI-GaMD simulations of barnase binding/dissociation to barstar. (*a*) Time courses of protein–protein interface distance calculated from six independent 2 μs PPI-GaMD simulations. (*b*) Original (reweighted) and modified (no reweighting) PMF profiles of the protein interface distance averaged over six PPI-GaMD simulations. Error bars are standard deviations of the free energy values calculated from six PPI-GaMD simulations. (*c*) 2D PMF profiles regarding the interface RMSD and the distance between the CZ atom of barnase Arg59 and CG atom of barstar Asp39. (*d*) 2D PMF profiles regarding the interface RMSD and the distance between the center of masses (COMs) of barnase residues Ala37-Ser38 and barstar residues Gly43-Trp44. (*e,f*) Low-energy conformations as identified from the 2D PMF profiles of the (*e*) intermediate "I1", (*f*) intermediate "I2". Strong electrostatic interactions are shown in red dash lines with their corresponding distance values labelled in the intermediate "I1" (*e*) and "I2" (*f*). Adapted with permission from Wang J, Miao Y. *Journal of Chemical Theory and Computation.* 10.1021/acs.jctc.1c00974. Copyright 2022 American Chemical Society.

≥3 binding and dissociation events as suggested by LiGaMD (Miao *et al.*, 2020), Pep-GaMD (Wang and Miao, 2020) and PPI GaMD (Wang and Miao, 2022) studies.

## Machine learning

Machine learning (ML) has been applied to improve computational docking, especially in the scoring functions (Khamis *et al.*, 2015). A scoring function in molecular docking refers to a

mathematical predictive model that outputs a representative score of the binding free energy of a bound conformation. Scoring of a docked complex is the final step of the three essential components in molecular docking, with the first two being chemical molecule representation and pose generation (Khamis *et al.*, 2015). A reliable scoring function should have a good scoring power (the ability to produce scores for different binding poses), ranking power (the ability to correctly rank a given set of ligands with known binding poses when bound to a common protein) and

docking power (the ability to identify the best binding pose of a given ligand from a set of computationally generated poses when bound to a specific protein; Ashtaway and Mahapatra, 2012). Kinnings *et al.* (2011) used a support vector machine (SVM) to derive a unique set of weights for each individual protein family – the $w_i$'s in the following equation:

$$\Delta G_{binding} = \frac{w_0 + w_1 \Delta G_{VdW} + w_2 \Delta G_{h-bond} + w_3 \Delta G_{rotor}}{+ w_4 \Delta G_{hydrophobic}} \quad (1)$$

This was shown to improve the binding affinity prediction of the electronic high throughput screening (eHiTS) molecular docking software (Zsoldos *et al.*, 2007) compared with empirical knowledge-based scoring functions (Khamis *et al.*, 2015). Similarly, a force field scoring function can be trained to derive a unique set of parameters for each individual protein family – the $A_{ij}$'s and $B_{ij}$'s in the following equation:

$$E_{binding} = \sum_{i=1}^{ligand} \sum_{j=1}^{protein} \left( \frac{A_{ij}}{r_{ij}^a} - \frac{B_{ij}}{r_{ij}^b} + 332 \frac{q_i q_j}{D r_{ij}} \right) \quad (2)$$

ML could also be used to predict the binding affinity based on a number of features of the protein–ligand complex, including geometric features, physical force field energy terms, pharmacophore features, etc. Specifically, ML could learn the relationship between these features and corresponding known binding affinity to predict the binding affinity of new complexes (Khamis *et al.*, 2015). Recently, Ballester and Mitchell (2010) applied non-parametric ML techniques to generate the functional form of scoring functions given molecular databases. The authors used random forest (RF; Breiman, 2001) to learn the relationship between the atomic-level description of the complex and the experimental binding affinity. Here, the $K_d$ and $K_i$ measurements were merged into a single binding constant $K$ to represent the experimental binding affinity. The atomic-level description used was of geometric nature and was the occurrence count of nine common elemental atoms (C, N, O, F, P, S, Cl, Br, I) type pair. Even though they completely neglected the energy terms induced by protein–ligand interactions, Ballester and Mitchell (2010) were able to achieve Pearson correlation coefficient of 0.774 on the PDBbind v2007 core set (195 complexes).

Very recently, deep learning (DL) methods, including RoseTTAFold (Baek *et al.*, 2021) and AlphaFold (Jumper *et al.*, 2021), were developed to achieve structure prediction accuracies far beyond those from classical force-field-based methods (Baek and Baker, 2022). These methods have millions of parameters, much more than the hundreds of parameters in classical approaches, thus better sample the large conformational space of proteins. Furthermore, they make no assumptions about the functional form of the interactions between atoms. In fact, the two DL-based methods learn millions of parameters directly to generate correct 3D structures from input amino acid sequences (Baek and Baker, 2022; Baek *et al.*, 2021; Jumper *et al.*, 2021). AlphaFold and RoseTTAFold are trained to predict structures from alignments of homologous amino acid sequences. In particular, the two DL-based approaches learn to extract rich structural information through a three-track network where information at the 1D sequence level, 2D distance map, and 3D coordinate level is successively transformed and integrated (Baek *et al.*, 2021; Jumper *et al.*, 2021). They were also shown to predict protein structures very accurately from single amino acid

sequences (Baek and Baker, 2022; Baek *et al.*, 2021; Jumper *et al.*, 2021).

MD simulations could generate very large data in terms of conformation frames and number of simulated atoms. For example, weighted ensemble of the COVID19 spike protein's closed-to-open state generated over 100 terabytes of data (Casalino *et al.*, 2021). This brings a challenge to identify proper CVs to differentiate conformational states from the raw simulation data and to identify corresponding biologically transitions between such states (e.g., open/closed states of spike). In this regard, the ML/deep learning has been applied to identify appropriate CV to analysis MD simulation trajectories (Noé, 2020; Wang *et al.*, 2020; Glielmo *et al.*, 2021; Sun *et al.*, 2022). These linear, non-linear and hybrid ML approaches cluster the simulation data along a small number of latent dimensions to identify conformational transitions between states (Bernetti *et al.*, 2020; Ramanathan *et al.*, 2012). Another benefit of MD-coupled ML approaches is that the information learned from ML can be used to iteratively guide the MD sampling (Wang *et al.*, 2019). Based on the predictive information bottleneck, Wang *et al.* (2019) developed an approach to identify system reaction coordinates and calculate the free energy and kinetic rates in biomolecules. The algorithm was demonstrated on conformational transitions in the alanine dipeptide model system and ligand dissociation from the L99A T4lysome. Thermodynamic and kinetic quantities calculated from short enhanced MD simulations for slow biomolecular processes were in good agreement with the experiments and long unbiased MD simulations.

Recently, we have integrated the GaMD, Deep Learning and free energy prOfiling Workflow (GLOW) to predict important reaction coordinates and map free energy profiles of biomolecules (Do *et al.*, 2022). First, GaMD simulations are performed on the target biomolecules (Fig. 4a). The residue contact map is then calculated for each GaMD simulation frame and transformed into images (Fig. 4b). The specialised type of neural network for image classification, two-dimensional (2D) convolutional neural network (CNN), is employed to classify the residue contact maps of target biomolecules, from which important residue contacts are identified by classic gradient-based pixel attribution (Fig. 4c). Finally, the free energy profiles of these reaction coordinates are calculated through reweighting of GaMD simulations to characterise the biomolecular systems of interest (Fig. 4d; Do *et al.*, 2022). GLOW was successfully demonstrated on characterisation of activation and allosteric modulation of a GPCR, using the adenosine $A_1$ receptor ($A_1AR$) as a model system. Characterisation of the $A_1AR$ activation was achieved by classification of the $A_1AR$ bound by "Antagonist", "Agonist" and "Agonist-Gi". GLOW achieved an overall accuracy of 99.34% and loss of 1.85%, respectively, on the validation data set after 15 epochs. Meanwhile, characterisation of $A_1AR$ allosteric modulation was achieved by classification of the $A_1AR$ bound by "Agonist-Gi" and "Agonist-Gi-PAM". GLOW achieved an overall accuracy of 99.27% and loss of 1.78%, respectively, on the validation data set after 15 epochs. GLOW identified characteristic residue contacts that were highly consistent with previous studies to the residue levels for both $A_1AR$ activation and allosteric modulation. In particular, the ligand-binding extracellular domains (ECL1–ECL3) and intracellular G-protein binding domains (TM3, TM5, TM6 and TM7) were found to be loosely coupled in the GPCR activation. Furthermore, it showed that ECL2 played a critical role in the allosteric modulation of $A_1AR$, being consistent with previous mutagenesis, structure and molecule modelling studies (Avlani *et al.*, 2007; Peeters *et al.*, 2012; Nguyen *et al.*, 2016; Miao *et al.*,

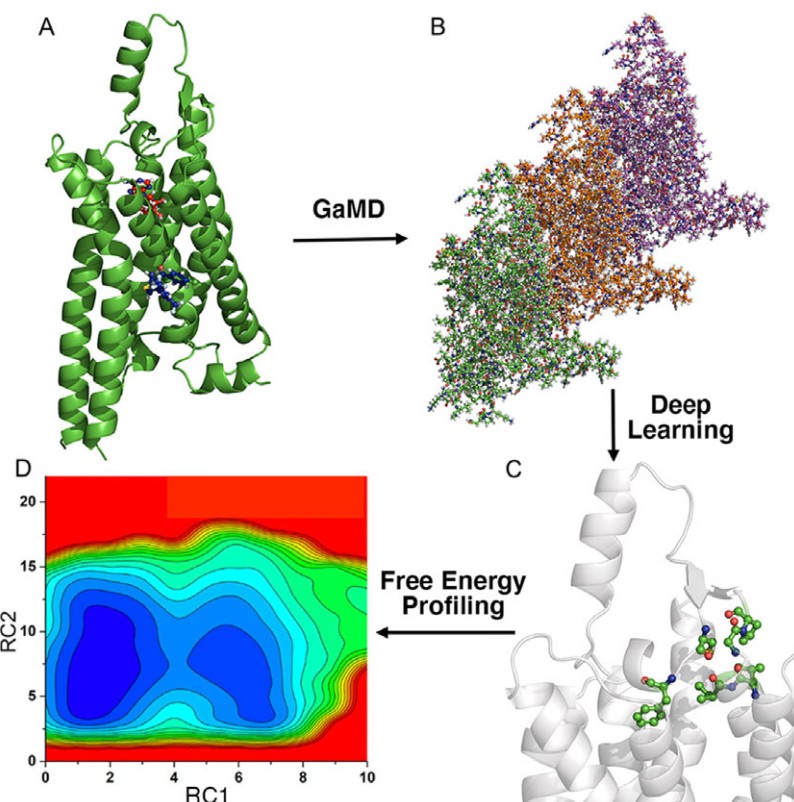

**Figure 4.** Overview of the Gaussian accelerated molecular dynamics (GaMD), deep learning (DL) and Free Energy PrOfiling Workflow (GLOW). (*a*) With structures of our interest, GaMD simulations are applied for enhanced sampling of the system dynamics. (*b*) DL models are then built with GaMD trajectories of residue contact maps transformed into image representations. (*c*) The DL analysis allows us to identify important residue contacts and system reaction coordinates (RCs). (*d*) Free energy profiles of the RCs are finally calculated through reweighting of GaMD simulations to characterise the system dynamics. Adapted with permission from Do HN, Wang J, Bhattari A and Miao Y. *Journal of Chemical Theory and Computation.* 10.1021/acs.jctc.1c01055. Copyright 2022 American Chemical Society.

2018a; Draper-Joyce *et al.*, 2021). GLOW revealed that binding of a PAM (MIPS521) to the agonist-Gi-$A_1$AR complex biased the receptor conformational ensemble, especially in the ECL1 and ECL2 regions. PAM binding stabilised agonist binding within the orthosteric pocket of $A_1$AR, which confined the extracellular mouth of the receptor Furthermore, PAM binding disrupted the N148$^{ECL2}$-V152$^{ECL2}$ α-helical hydrogen bond and distorted this portion of the ECL2 helix (Do *et al.*, 2022).

In addition, DL has been widely applied to optimise force field (Poltavsky and Tkatchenko, 2021; Unke *et al.*, 2021; Chatterjee *et al.*, 2022), binding free energy calculations (Jiang *et al.*, 2021; Jones *et al.*, 2021; Chen *et al.*, 2022) and binding pathway identification (Motta *et al.*, 2022).

### Conclusions and outlook

With remarkable advances in both computer hardware and software, computational approaches have achieved significant improvement to characterise biomolecular recognition, including molecular docking, MD simulations and ML. ML has been incorporated into both molecular docking and MD simulations to improve the docking accuracy, simulation efficiency and trajectory analysis, e.g., AlphaFold-Multimer and GLOW. MD simulations have enabled characterisation of biomolecular binding thermodynamics and kinetics, attracting increasing attention in recent years. Long time scale cMD simulations have successfully captured

biomolecular binding processes, although slow dissociation of biomolecules are still often difficult to simulate using cMD.

Enhanced sampling methods have greatly reduced the computational cost for calculations of biomolecular binding thermodynamics and kinetics. Higher sampling efficiency could be generally obtained using the CV-based methods than using the CV-free methods. However, CV-based enhanced sampling methods require predefined CVs, which is often challenging for simulations of complex biological systems. Nevertheless, ML techniques have proven useful to identify proper CVs or reaction coordinates. Alternatively, CV-free methods are usually easy to use without requirement of *a priori* knowledge of the studied systems. Additionally, the CV-based and CV-free methods could be combined to be more powerful. The CV-free methods can enhance the sampling to potentially overcome the hidden energy barriers in orthogonal degrees of freedom relative to the CVs predefined in the CV-based methods, which could enable faster convergence of the MD simulations. Newly developed algorithms in this direction include integration of replica exchange umbrella sampling with GaMD (GaREUS; Oshima *et al.*, 2019), replica exchange of solute tempering with umbrella sampling (gREST/ REUS; Kamiya and Sugita, 2018; Re *et al.*, 2019), replica exchange of solute tempering with well-tempered Metadynamics (ST-MetaD; Mlýnský *et al.*, 2022) and temperature accelerated molecular dynamics (TAMD) with integrated tempering sampling (ITS/TAMD; Xie *et al.*, 2017).

Recent years have seen an increasing number of techniques that introduce "selective" boost in the CV-free enhanced sampling methods, including the selective ITS, selective scaled MD, selective aMD and selective LiGaMD, Pep-GaMD and PPI-GaMD. In these methods, only essential energy terms are selectively boosted to further increase the sampling efficiency. Additionally, compatible enhanced sampling methods could be combined to be more powerful. For example, GaMD has been combined with Umbrella Sampling to achieve significantly improved efficiency (Oshima *et al.*, 2019; Wang *et al.*, 2021). Besides enhanced sampling, the accuracy of force fields and water models play a critical role in predicting the biomolecular binding affinities and kinetics. For example, the TIP4P2015 water model was shown to be more accurate than the TIP3P water model in calculating the kinetics of barnase–barstar binding in cMD simulations (Pan *et al.*, 2019). Nevertheless, biomolecular recognition in systems of increasing sizes (such as viruses and cells) and accurate calculations of binding thermodynamics and kinetics of large biomolecular complexes present grand challenges for computational modelling and enhanced sampling simulations. Further innovations in both computing hardware and method developments may help us to address these challenges in the future.

**Acknowledgements.** This work used supercomputing resources with allocation awards TG-MCB180049 and BIO210039 through the Extreme Science and Engineering Discovery Environment (XSEDE), which is supported by National Science Foundation Grant no. ACI-1548562 and project M2874 through the National Energy Research Scientific Computing Center (NERSC), which is a U.S. Department of Energy Office of Science User Facility operated under Contract No. DE-AC02-05CH11231. It also used computational resources provided by the Research Computing Cluster at the University of Kansas. This work was supported in part by the National Institutes of Health (R01GM132572), National Science Foundation (2121063) and the startup funding in the College of Liberal Arts and Sciences at the University of Kansas.

**Open Peer Review.** To view the open peer review materials for this article, please visit http://doi.org/10.1017/qrd.2022.11.

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
