## [Reviewer Report]

*Comments to Author*: In this manuscript, the authors introduce recent enhanced-sampling methods for accelerating association and dissociation events of protein-ligand, protein-peptide, and protein-protein complexes. The authors classify the method into the three types: Collective-variable (CV) based methods, CV-free methods, and the methods combined with machine learning (ML) techniques. In CV-based methods, bias potentials are applied to the system along the predefined CVs. Umbrella sampling or metadynamics are applied to binding problems to investigate binding affinities, pathways, and kinetics. In CV-free methods, bias potentials do not depend on the CVs. The authors mainly introduce Gaussian accelerated MD (GaMD), which was developed by themselves. In particular, selective GaMD methods are efficient for binding and unbinding simulations, because they can apply the boosting potentials to the selective regions of interest in the system. Due to sufficient statistics for binding and unbinding events, the free-energy changes as well as the kinetics (k_on and k_off) can be estimated with high accuracy. In the methods with ML, ML or deep learning (DL) improves the scoring function for docking simulations and achieves the structure prediction, such as AlphaFold and RoseTTAFold. The authors also combine DL with GaMD. DL extracts the important interactions between residues and the CVs from GaMD trajectories, which enables to obtain the accurate free-energy profiles. This manuscript is well written and concisely summarizes recent works of enhanced sampling methods. I recommend the publication of this manuscript after minor revisions, considering the points below.

The authors separately discuss about CV-based and CV-free methods, but their combination should be important for more efficient sampling. In fact, in the last paragraph of Sec.5, the authors mention that compatible enhanced methods could be combined to be more powerful. Even if the hidden energy barriers exist in the orthogonal degrees of freedom for the predefined CVs in the CV-based method, the CV-free method can enhance the sampling in the orthogonal CV spaces. Several combinations of CV-based and CV-free methods have been already proposed. For examples, GaREUS (https://doi.org/10.1021/acs.jctc.9b00761), gREST/REUS (https://doi.org/10.1063/1.5016222; https://doi.org/10.1073/pnas.1904707116), ST-MetaD (https://doi.org/10.1021/acs.jctc.1c01222), ITS/TAMD (https://doi.org/10.1063/1.4973607), etc. The authors should discuss more about the combinations of enhanced sampling methods.

GaMD boosts the motion and flexibility of biomolecules and enhances the sampling in the conformational space, resulting in the reduction of the simulation time. However, even if GaMD is used, many independent GaMD simulations or long GaMD simulations are required to obtain sufficient statistics for protein-peptide binding or binding between large biomolecules. We suggest the authors to discuss convergence issues of GaMD in more details.

GaMD successfully reproduces the binding affinities and kinetics with very high accuracy. However, even if the binding and unbinding events are sufficiently sampled, the affinities and kinetics would strongly depend on the force-field parameters of proteins and ligands and the water model. The author had better explain the relationship between the force-field parameters and enhanced conformational sampling methods.

Minor comments

1. Page 11, Fifth paragraph of Section 3: V_{PP,nb}(r_P) + V_{LL,nb}(r_L) + V_{EE,nb}(r_E) duplicates in V(r). Please modify the duplication.

---

## [Reviewer Report]

*Comments to Author*: The manuscript reviews computational approaches to study biomolecular binding and dissociation processes. The authors reviewed the challenges and latest developments in applying molecular dynamics (MD) simulations to study protein-ligand and protein-protein interactions. Among these methods, they described in detail how Gaussian accelerated MD (GaMD) can be used to enhance sampling. A "selective GaMD" algorithm was introduced to more efficiently accelerate a certain biological process by perturbing specific terms in the potential energy function. In general this review is very interesting to the readership of QRB discovery - I would like to recommend it for publication after considering the following minor points:

1. On Page 8, it is mentioned that metadynamics simulations were used to predict ligand unbinding pathways and related k_off. "The predicted k_off (9.1 ± 2.5 s^-1) was comparable with the experimental data (600 ± 300 s^-1)." Actually these two numbers are not exactly comparable as they are orders of magnitude away from each other.

2. The authors discussed coarse-grained (CG) MD models, which can greatly extend the simulation timescales compared to conventional MD. They should also address CG models that can efficiently sample peptide binding to a receptor. A useful united-atom CG model (PACE) was successfully used to study intrinsically disordered peptide binding to a receptor (Han, W., & Schulten, K. (2014). JACS, 136(35), 12450-12460). This work performed millisecond CG simulations to characterize an Aβ peptide binding to an amyloid fibril tip.

3. The original work of milestoning should be cited in the discussion of SEEKR on Page 6 (e.g. a review by Elber, R. (2020). Annu. Rev. Biophys., 49(1), 69-85).

4. In Figure. 4B, it is a bit unclear to me what the multiple structures represent. Are these structures just static PDB snapshots from GaMD or should they be the saliency maps built based on residue contacts?

5. One of the benefits of MD-coupled machine learning approaches is that the information (features) learned from the neural network can be used to iteratively enhance the MD sampling. This point can be discussed in Section 4 (e.g. check out Wang, Y., Ribeiro, J.M.L. & Tiwary, P. (2019). Nat. Commun., 10, 3573).

---

## [Reviewer Report]

*Comments to Author*: Reviewer #1: The manuscript reviews computational approaches to study biomolecular binding and dissociation processes. The authors reviewed the challenges and latest developments in applying molecular dynamics (MD) simulations to study protein-ligand and protein-protein interactions. Among these methods, they described in detail how Gaussian accelerated MD (GaMD) can be used to enhance sampling. A "selective GaMD" algorithm was introduced to more efficiently accelerate a certain biological process by perturbing specific terms in the potential energy function. In general this review is very interesting to the readership of QRB discovery - I would like to recommend it for publication after considering the following minor points:

1. On Page 8, it is mentioned that metadynamics simulations were used to predict ligand unbinding pathways and related k_off. "The predicted k_off (9.1 ± 2.5 s^-1) was comparable with the experimental data (600 ± 300 s^-1)." Actually these two numbers are not exactly comparable as they are orders of magnitude away from each other.

2. The authors discussed coarse-grained (CG) MD models, which can greatly extend the simulation timescales compared to conventional MD. They should also address CG models that can efficiently sample peptide binding to a receptor. A useful united-atom CG model (PACE) was successfully used to study intrinsically disordered peptide binding to a receptor (Han, W., & Schulten, K. (2014). JACS, 136(35), 12450-12460). This work performed millisecond CG simulations to characterize an Aβ peptide binding to an amyloid fibril tip.

3. The original work of milestoning should be cited in the discussion of SEEKR on Page 6 (e.g. a review by Elber, R. (2020). Annu. Rev. Biophys., 49(1), 69-85).

4. In Figure. 4B, it is a bit unclear to me what the multiple structures represent. Are these structures just static PDB snapshots from GaMD or should they be the saliency maps built based on residue contacts?

5. One of the benefits of MD-coupled machine learning approaches is that the information (features) learned from the neural network can be used to iteratively enhance the MD sampling. This point can be discussed in Section 4 (e.g. check out Wang, Y., Ribeiro, J.M.L. & Tiwary, P. (2019). Nat. Commun., 10, 3573).

Reviewer #2: In this manuscript, the authors introduce recent enhanced-sampling methods for accelerating association and dissociation events of protein-ligand, protein-peptide, and protein-protein complexes. The authors classify the method into the three types: Collective-variable (CV) based methods, CV-free methods, and the methods combined with machine learning (ML) techniques. In CV-based methods, bias potentials are applied to the system along the predefined CVs. Umbrella sampling or metadynamics are applied to binding problems to investigate binding affinities, pathways, and kinetics. In CV-free methods, bias potentials do not depend on the CVs. The authors mainly introduce Gaussian accelerated MD (GaMD), which was developed by themselves. In particular, selective GaMD methods are efficient for binding and unbinding simulations, because they can apply the boosting potentials to the selective regions of interest in the system. Due to sufficient statistics for binding and unbinding events, the free-energy changes as well as the kinetics (k_on and k_off) can be estimated with high accuracy. In the methods with ML, ML or deep learning (DL) improves the scoring function for docking simulations and achieves the structure prediction, such as AlphaFold and RoseTTAFold. The authors also combine DL with GaMD. DL extracts the important interactions between residues and the CVs from GaMD trajectories, which enables to obtain the accurate free-energy profiles. This manuscript is well written and concisely summarizes recent works of enhanced sampling methods. I recommend the publication of this manuscript after minor revisions, considering the points below.

The authors separately discuss about CV-based and CV-free methods, but their combination should be important for more efficient sampling. In fact, in the last paragraph of Sec.5, the authors mention that compatible enhanced methods could be combined to be more powerful. Even if the hidden energy barriers exist in the orthogonal degrees of freedom for the predefined CVs in the CV-based method, the CV-free method can enhance the sampling in the orthogonal CV spaces. Several combinations of CV-based and CV-free methods have been already proposed. For examples, GaREUS (https://doi.org/10.1021/acs.jctc.9b00761), gREST/REUS (https://doi.org/10.1063/1.5016222; https://doi.org/10.1073/pnas.1904707116), ST-MetaD (https://doi.org/10.1021/acs.jctc.1c01222), ITS/TAMD (https://doi.org/10.1063/1.4973607), etc. The authors should discuss more about the combinations of enhanced sampling methods.

GaMD boosts the motion and flexibility of biomolecules and enhances the sampling in the conformational space, resulting in the reduction of the simulation time. However, even if GaMD is used, many independent GaMD simulations or long GaMD simulations are required to obtain sufficient statistics for protein-peptide binding or binding between large biomolecules. We suggest the authors to discuss convergence issues of GaMD in more details.

GaMD successfully reproduces the binding affinities and kinetics with very high accuracy. However, even if the binding and unbinding events are sufficiently sampled, the affinities and kinetics would strongly depend on the force-field parameters of proteins and ligands and the water model. The author had better explain the relationship between the force-field parameters and enhanced conformational sampling methods.

Minor comments

1. Page 11, Fifth paragraph of Section 3: V_{PP,nb}(r_P) + V_{LL,nb}(r_L) + V_{EE,nb}(r_E) duplicates in V(r). Please modify the duplication.